# Remaining Useful Life Prediction of Aircraft Turbofan Engine Based on Random Forest Feature Selection and Multi-Layer Perceptron

Hairui Wang [1], Dongwen Li [1], Dongjun Li [1], Cuiqin Liu [1], Xiuqi Yang [2] and Guifu Zhu [3,*]

[1] Faculty of Information Engineering and Automation, Kunming University of Science and Technology, Kunming 650504, China
[2] School of Materials Science and Engineering, Kunming University of Science and Technology, Kunming 650504, China
[3] Information Technology Construction Management Center, Kunming University of Science and Technology, Kunming 650504, China
* Correspondence: zhuguifu@kust.edu.cn

**Abstract:** The accurate prediction of the remaining useful life (RUL) of aircraft engines is crucial for improving engine safety and reducing maintenance costs. To tackle the complex issues of nonlinearity, high dimensionality, and difficult-to-model degradation processes in aircraft engine monitoring parameters, a new method for predicting the RUL of aircraft engines based on the random forest algorithm and a Bayes-optimized multilayer perceptron (MLP) was proposed here. First, the random forest algorithm was used to evaluate the importance of historical monitoring parameters of the engine, selecting the key features that significantly impact the engine's lifetime operation cycle. Then, the single exponent smoothing (SES) algorithm was introduced for smoothing the extracted features to reduce the interference of original noise. Next, an MLP-based RUL prediction model was established using a neural network. The Bayes' online parameter updating formula was used to solve the objective function and return the optimal parameters of the MLP training model and the minimum value of the evaluation index RMSE. Finally, the probability density function of the predicted RUL value of the aircraft engine was calculated to obtain the RUL prediction results. The effectiveness of the proposed method was verified and analyzed using the C-MAPSS dataset for turbofan engines. Experimental results show that, compared with several other methods, the RMSE of the proposed method in the FD001 test set decreases by 6.1%, demonstrating that the method can effectively improve the accuracy of RUL prediction for aircraft engines.

**Keywords:** aeroengine; random forest; Bayesian parameter updating; MLP; probability density function; remaining useful life

## 1. Introduction

Aircraft engines are essential components of aviation equipment and directly impact flight conditions and the safety performance of aircraft [1]. To ensure an optimal engine performance, prognostics and health management technology is widely used for engine predictive maintenance. This method evaluates the health and remaining useful life (RUL) of the equipment using historical data and provides maintenance predictions to enhance equipment reliability and safety [2].

As one of the cores of the prognostics and health management (PHM) method, aero-engine RUL prediction has been divided into three main approaches by other scholars: physical-model-based methods [3], data-driven methods [4], and hybrid methods [5]. The physical-model-based approach mainly relies on the engine's internal structure to construct a physical model, which has a high uncertainty and increases the prediction difficulty. The data-driven approach uses machine learning tools to mine the implicit relationship between

historical monitoring data and RUL to construct a prediction model. The hybrid approach combines both methods, mixing their advantages and disadvantages. Thus, data-driven approaches are currently a hot research topic.

While deep learning methods offer some advantages over traditional methods in predicting the RUL of aerospace engines, additional research is needed to explore their potential for fault maintenance and health assessment. Bingce et al. [6] estimated the RUL prediction interval of an aeroengine using a multilayer perceptron (MLP) model based on information granularity theory. Guoxing et al. [7] used the Euclidean distance method to identify the initial lifetime. Qiyao et al. [8] fused the time-series of different equipment sensors with MLP prediction models. Jun et al. [9] utilized deep learning techniques to extract the sensor time-series signals between hidden dependencies and adapted the DLSTM network structure and parameters with an adaptive moment estimation algorithm, which showed a good performance compared to other neural network models.

The historical degradation data of aerospace engines are characterized by a high dimensionality and massive scale. Given the size of the data, the current research focus is on identifying the most valuable information. In terms of feature screening and extraction, Wennian et al. [10] utilized a recurrent neural network to convert high-dimensional data monitored by sensors into low-dimensional data, constructing a one-dimensional health index (HI). Cunsong et al. [11] employed the k-means algorithm to extract degradation features with monotonic trends and employed a deep forest classifier with LSTM to create an RUL degradation trend prediction model. Zhang et al. [12] proposed a method for predicting the remaining useful life of engines using a combination of convolutional and recurrent neural networks (CNN-RNN), and the prediction accuracy was further improved by processing and clustering the data.

RUL prediction for aero engines using similarity matching methods is a current hot research topic. In their study, Lam et al. [13] evaluated the trajectory similarity prediction (TSBP) method using similarity regression (SLR) based on Pearson correlation coefficients with dynamic time warping (DTW) methods to determine the most similar degradation model to the test data. They used the principle of weighted kernel density estimation to quantify the RUL prediction. Cai et al. [14] used the kernel two-sample test (KTST) similarity matching procedure by fitting a distribution to the RUL prediction results from a historical database of sample data from operating cycles to failure to obtain confidence intervals for RUL predictions. However, most of the similarity matching was performed using a single time scale for the construction of health indicators, resulting in obtaining degradation trajectories with inconsistent lengths. This inconsistency is not conducive to the construction of historical engine degradation trajectories.

To address the complex problems of a high dimensionality, non-linearity, and difficulty in establishing the historical decline process of aero-engine monitoring parameters, this paper proposes an MLP aero-engine remaining useful life prediction method based on random forest and Bayesian updating. The random forest algorithm was introduced to analyze and evaluate the degree of influence of each monitoring feature on the remaining life cycle of the engine and calculate the corresponding weights. This was carried out to effectively solve problems such as the large number of engine monitoring parameter features and the difficulty in extracting them, and to eliminate the features with the worst weight score values. Extracted parameters were exponentially smoothed to reduce noise interference. An RUL training model for MLP engines based on Bayesian parameter updating was then constructed, and the remaining service life probability density distribution of the aeroengine was derived. The expectation and standard deviation of the RUL prediction were obtained. The method proposed in this paper was validated using NASA's C-MAPPS dataset, and the RUL prediction results were evaluated using the RMSE and $R^2$. The results show that the accuracy of the engine RUL prediction can be effectively improved compared with other methods. Our methodology has been shown to significantly enhance the accuracy of RUL prediction for engines.

## 2. RUL Prediction Methods

### 2.1. Random Forest Algorithm

There are many factors that affect the life cycle of an aeroengine, and the extent to which each factor affects its life cycle varies. Therefore, it is necessary to explore and analyze the degree of influence of each factor on the life cycle of an aeroengine. The random forest algorithm has shown good performance in the parameter selection process, making it suitable for high-dimensional data and effectively avoiding the problem of a weak generalization ability in training models. Random forest is an integrated learning algorithm with decision trees as the base learner, and solves the performance bottleneck problem of decision trees [15]. Random forest uses a bootstrapping method for model construction. It randomly selects a sample set of size *m* with a replacement from the original training set and performs *n* iterations to generate *n* training sets, which can train *n* decision tree models. The nodes are split at each iteration based on the Gini coefficient or information gain to select the best features for splitting. The importance of the features is measured by averaging the decline in the residuals of each feature over each decision tree. The Gini index is used to obtain the importance of a feature's contribution to the overall decision tree node value. Assuming that there are *J* features, *I* decision trees, and *C* categories, the Gini index for the *i*th tree node is calculated as follows:

$$GI_q^{(i)} = 1 - \sum_{c=1}^{|C|} \left( p_{qc}^{(i)} \right)^2, \tag{1}$$

where $p_{qc}$ represents the proportion of category *c* in node *q*. The more the value of *GI* decreases, the higher the relative importance of the feature. The change in Gini index before and after the node splitting is:

$$Y_{jq}^{(Gini)(i)} = GI_q^{(i)} - GI_I^{(i)} - GI_r^{(i)}, \tag{2}$$

The importance score (*Y*) of $x_i$ in the *i*th tree is:

$$Y_j^{(Gini)} = \sum_{i=1}^{I} Y_i^{(Gini)(i)}, \tag{3}$$

The process of feature extraction in random forests is illustrated in Figure 1.

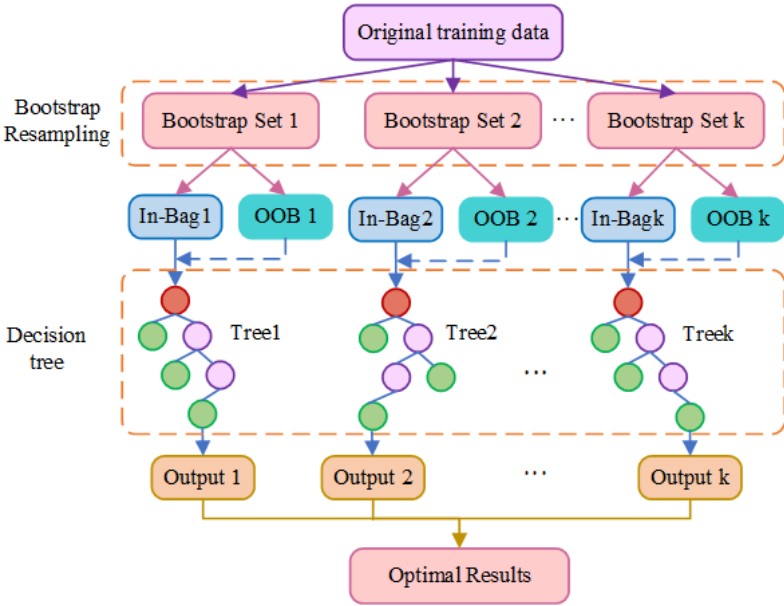

**Figure 1.** Random forest feature extraction model.

### 2.2. Exponential Smoothing

Due to the significant noise in the monitoring features of the C-MPSS dataset, the simple exponential smoothing (SES) algorithm is introduced to smooth the raw monitoring features of the engine. This is carried out to enable the prediction algorithm model to better capture the changing trend of the feature signal. The current actual value of the engine sensor monitoring parameter is weighted averaged with the previous parameter value to obtain the smoothing result [16]. The formula for the calculation is:

$$\begin{cases} y_t = x_t, t = 1 \\ y_t = \alpha x_t + (1-\alpha)y_{t-1}, t \geq 2, \end{cases} \tag{4}$$

In the equation, $x_t$ represents the true value at time $t$ and $y_{t-1}$ represents the observed value at time $t-1$. $\alpha \in (0,1)$ is the decay factor. The smoothing parameter $\alpha$ controls the rate of the weight reduction of the exponential smoothing. When the value of $\alpha$ is closer to 1, the recent characteristic parameters have a greater influence on the prediction results; when $\alpha$ is closer to 0, the data parameters are smoother.

### 2.3. Multilayer Perception Machine

A multilayer perceptron is a type of artificial neural network (ANN). The multilayer perceptron consists of an input layer, an output layer, and multiple hidden layers [17,18]. The neural network structure with three layers of perception machines is displayed in Figure 2, where each circle represents a neural node. The input layer receives raw input signals from external sources, and the hidden and output layers process these signals to produce the final output.

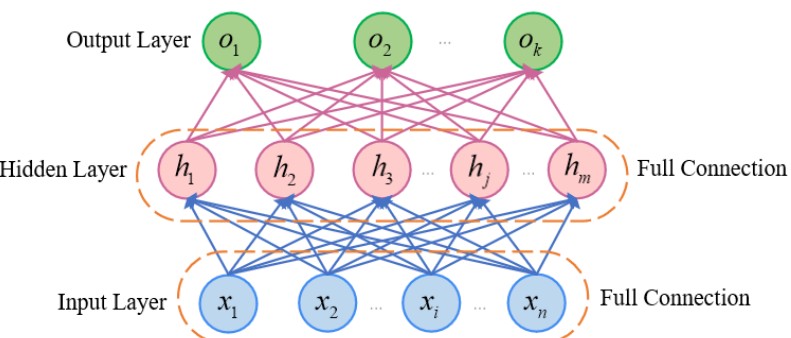

**Figure 2.** MLP structure diagram.

The input layer of the MLP is denoted by $X \in R^{m \times n}$, with $m$ samples and $n$ features. If the MLP has a single hidden layer, the weights of the hidden layer are represented by $W_h \in R^{n \times h}$ and the bias parameters are denoted by $b_h \in R^{1 \times h}$. The output of the hidden layer is calculated as follows:

$$H = \sigma(XW_h + b_h), \tag{5}$$

If there are $q$ label values in the output, the weight and bias parameters for the output layer are denoted by $W_o \in R^{h \times q}$ and $b_h \in R^{1 \times q}$, respectively. The formula for the output layer is:

$$O = HW_o + b_o, \tag{6}$$

### 2.4. Bayesian Parameter Update Algorithm

The Bayesian optimization algorithm builds a collection function based on the posterior probability distribution model and utilizes previous evaluation information to select the next optimal parameter evaluation point. As the degradation process of engine monitoring parameters is stochastic and variable, this paper employed the Gaussian process in

the Bayesian algorithm with an acquisition function to optimize the stochastic problem of the MLP in remaining life prediction.

Specifically, the Gaussian process (GP) models the probability distribution of the function.

$$f(\vec{x}) \sim N(\mu(x), k(x, x)), \tag{7}$$

Here, $x$ represents the training data for the Gaussian process, which follow a multivariate Gaussian distribution for all $\vec{x} = [x_1, x_2, \cdots, x_n]$. The covariance function is denoted by $K(x, x')$ and the sample mean is denoted by $u(x)$.

The optimal solution is obtained by evaluating sample candidates using the acquisition function, expressed as:

$$f(x) = \varphi\left(\frac{\mu(x) - f(x_{\max}) - \alpha}{\sigma(x)}\right), \tag{8}$$

The equation includes $\sigma(x)$, which is the variance of the Gaussian process, $\alpha$ being the hyperparameter used to search for the optimal output in the model. Figure 3 illustrates the flow of the MLP prediction model based on Bayes optimization.

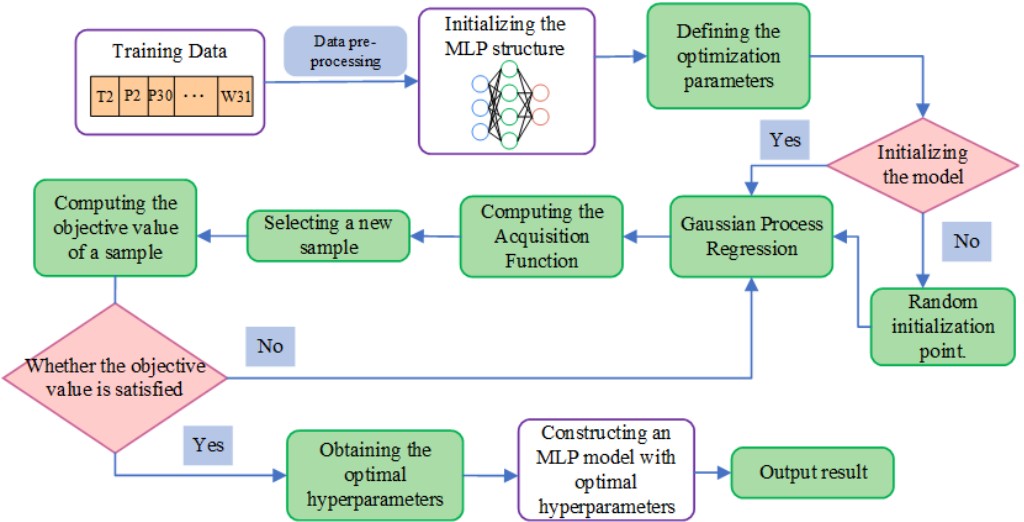

**Figure 3.** Bayesian flowchart for optimizing MLP parameters.

*2.5. Probability Density Function (PDF) of Remaining Life*

Assuming that the RUL prediction of the target aeroengine after model training at operating cycle $T = \{t_1, t_2, \cdots, t_k\}$ is $X = \{x_{t_1}, x_{t_2}, \cdots, x_{t_k}\}$, we can calculate its expected value and standard deviation as follows:

$$\mu_{x_{t_k}} = \frac{1}{K}\sum_{i=1}^{K} x_{t_i}, \tag{9}$$

$$\sigma_{x_{t_k}} = \left(\frac{1}{K}\sum_{i=1}^{K}\left(x_{t_i} - \mu_{x_{t_k}}\right)^2\right)^{\frac{1}{2}}, \tag{10}$$

Under a Gaussian distribution, the PDF for the remaining engine life $x_{t_k}$, with an expectation of $\mu_{x_{t_k}}$ and a standard deviation of $\sigma_{x_{t_k}}$ at time t of the current operating cycle, is:

$$f\left(x_{t_k}\right) = \frac{1}{\sigma_{x_{t_k}}\sqrt{2\pi}}e^{-\frac{\left(X - \mu_{x_{t_k}}\right)^2}{2\sigma_{x_{t_k}}{}^2}}, \tag{11}$$

When the standard deviation is larger, the peak is lower and the prediction uncertainty is reduced. Using the above formula, we can determine the PDF and mathematical expectation of the engine's remaining life at any operating cycle, enabling an accurate prediction of the RUL.

## 3. RF and Bayesian MLP-Based RUL Prediction Framework

Step 1: Data preprocessing.

An aero-engine's monitoring parameters contain a large amount of historical degradation data. To enhance the prediction performance of the training model, it is important to select monitoring parameters that strongly correlate with the engine's life cycle. We employed a random forest algorithm to generate an importance ranking value for each monitoring parameter, and subsequently eliminated parameters with low importance. The SES algorithm was used to smooth the extracted features and reduce the influence of noise, allowing samples to retain the maximum amount of degradation information. As monitoring parameters mostly consist of different types of data, we employed the MinMax scaling method to normalize the monitoring features and reduce the influence of data magnitude.

Step 2: Training the multi-layer perceptron model.

Our training framework is based on artificial neural networks. Firstly, we filtered the parameters that require updating and determined the range of the search, with the root-mean-square error (RMSE) as our search target. We obtained the best possible parameters, which were then employed as the final combinations for the MLP engine RUL prediction model.

Step 3: RUL prediction.

We began by pre-processing the test set, which was then input into the Bayes MLP prediction model for training. The prediction evaluation indexes RMSE and $R^2$ values were output, and we compared and analyzed the prediction results with the actual values. Next, we calculated the expectation and standard deviation of the RUL prediction values in order to obtain the PDF of the engine's RUL. A diagram of the overall RUL prediction process framework for aero engines can be found in Figure 4.

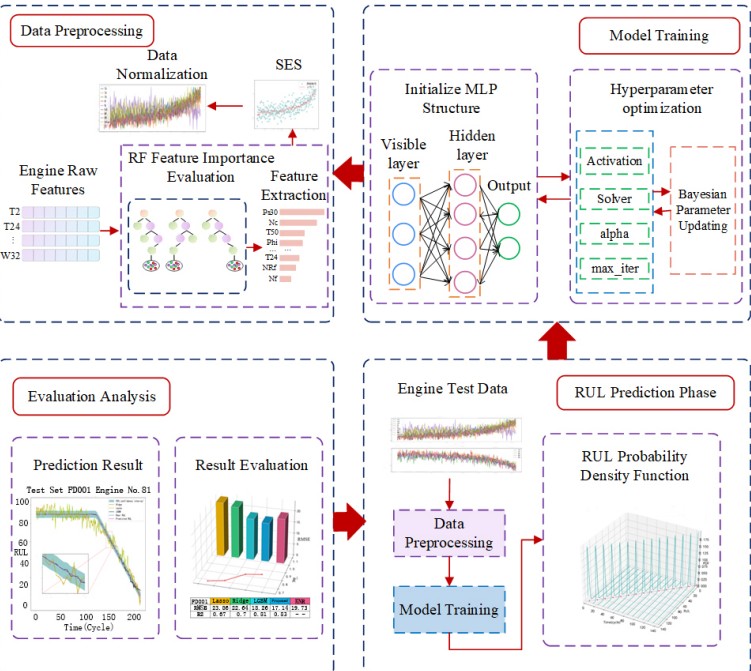

**Figure 4.** Flowchart for RUL prediction of aircraft engines.

## 4. Experimental Validation and Analysis

### 4.1. Validation Data Set

The validation data are based on the CMAPSS dataset provided by the NASA Ames Failure Prediction Research Center. It simulates the health of each output component during the degradation of a real commercial turbofan engine in different operating environments [19]. The dataset consists of four training and test sets, FD001 to FD004, which were simulated under various operating conditions and failure modes [20]. The training dataset comprises the full life cycle data from the start of the aircraft's in-service operation until failure, whereas the test set records the partial life cycle data of the aircraft from its initial state until failure [21,22]. The dataset was subjected to a degree of engine wear during the simulation due to interference from artificial noise. The size of the dataset is presented in Table 1:

**Table 1.** C-MAPSS dataset.

| Dataset | FD001 | FD002 | FD003 | FD004 |
|---|---|---|---|---|
| Number of Training Samples | 100 | 260 | 100 | 249 |
| Number of Test Samples | 100 | 259 | 100 | 248 |
| Total Length of Training Set | 20,630 | 53,758 | 24,719 | 61,248 |
| Total Length of Test Set | 13,095 | 33,990 | 16,595 | 41,213 |
| Operating Conditions | 1 | 6 | 1 | 6 |
| Fault Mode | 1 | 1 | 2 | 2 |

Table 2 presents the specific meaning of each monitoring parameter in the dataset. We note that, within the dataset, Setting_1, Setting_2 and TRA correspond to three different sets of operating environment conditions.

**Table 2.** Description of Monitoring Parameters.

| No. | Parameters | Unit |
|---|---|---|
| 1 | Altitude (Setting_1) | – |
| 2 | Mach number (Setting_2) | – |
| 3 | Throttle resolver angle (TRA) | – |
| 4 | Total temperature at fan inlet (T2) | °R |
| 5 | LPC outlet temperature (T24) | °R |
| 6 | HPC outlet temperature (T30) | °R |
| 7 | LPT outlet temperature (T50) | °R |
| 8 | Fan inlet pressure (P2) | psia |
| 9 | Bypass duct pressure (P15) | psia |
| 10 | HPC outlet pressure (P30) | psia |
| 11 | Physical fan speed (Nf) | r/min |
| 12 | Physical core speed (Nc) | r/min |
| 13 | Engine pressure ratio P50/P2 (Epr) | – |
| 14 | HPC outlet static pressure (Ps30) | psia |
| 15 | Ratio of fuel flow to Ps30 (Phi) | pps/psia |
| 16 | Corrected fan speed (NRf) | r/min |
| 17 | Corrected core speed (NRc) | r/min |
| 18 | Corrected core speed (BPR) | – |
| 19 | Burner fuel–air ratio (FarB) | – |
| 20 | Bleed enthalpy (htBleed) | – |
| 21 | Required fan speed (Nf_dmd) | r/min |
| 22 | Required fan conversion speed (PCNfR_dmd) | r/min |
| 23 | High-pressure turbines cool air flow (W31) | lbm/s |
| 24 | Low-pressure turbines cool air flow (W32) | lbm/s |

The maximum cycle sample length distribution of all engines in datasets FD001-FD004 is shown in Figure 5. The sample lengths of the four datasets vary. FD001 and FD003 datasets were collected under a single operating condition, with sample lengths mainly

distributed between [150 and 250], whereas datasets FD002 and FD004 were collected under six operating conditions and two fault modes, with sample lengths mainly distributed between [150 and 300]. This indicates that the engine's life cycle is shorter under a single operating condition, resulting in a large amount of degradation trajectory information that cannot be utilized.

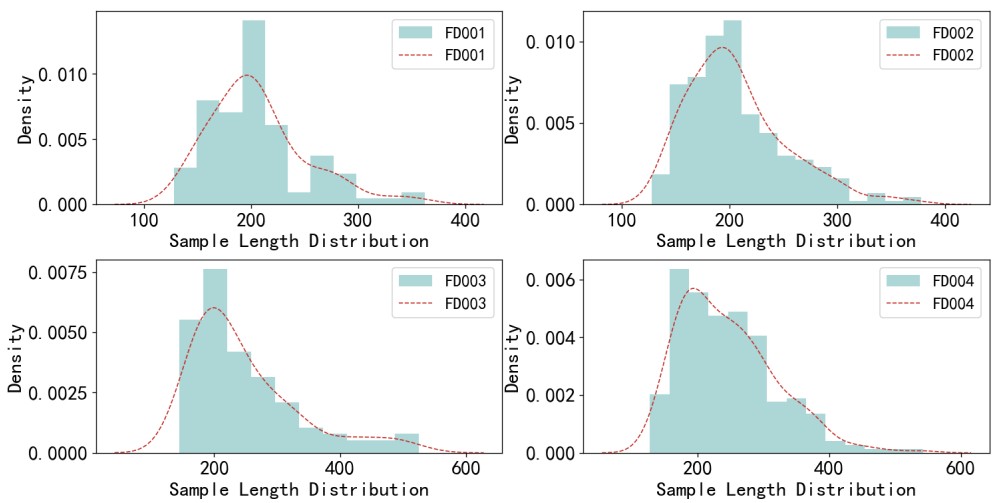

**Figure 5.** Distribution of data sample lengths.

*4.2. Evaluation Metrics*

To measure the effectiveness of the prediction model proposed, evaluation indicators, including root mean square error, mean absolute error, coefficient of determination ($R^2$), and performance score function (Score), were introduced to assess the model's performance from four dimensions. The calculation method for the RMSE [23] is as follows:

$$\text{RMSE} = \sqrt{\frac{1}{n}\sum_{i=1}^{n}(\hat{y}_i - y_i)^2}, \tag{12}$$

The RMSE penalizes overestimation and underestimation errors in predicting the RUL of the engine equally, whereas Score [24] penalizes these errors differently. Therefore, to enhance the accuracy of model evaluation, the performance score function was introduced.

$$\text{Score} = \begin{cases} \sum_{i=1}^{n}\left(e^{\frac{-(\hat{y}_i - y_i)}{13}} - 1\right), \hat{y}_i - y_i < 0 \\ \sum_{i=1}^{n}\left(e^{\frac{\hat{y}_i - y_i}{10}} - 1\right), \hat{y}_i - y_i > 0 \end{cases}, \tag{13}$$

The mean absolute error (MAE) [25] represents the dispersion level among data samples, and its calculation formula is as follows:

$$\text{MAE} = \frac{1}{n}\sum_{i}^{n}|\hat{y}_i - y_i|, \tag{14}$$

The smaller the predicted values for the RMSE, MAE, and Score, the better the model's predictive performance. For the $R^2$, the closer its value is to 1, the better the fitting degree of the prediction model.

$$R^2 = 1 - \frac{\sum_{i=1}^{n}(y_i - \hat{y}_i)^2}{\sum_{i=1}^{n}(y_i - \overline{y_i})^2}, R \in [0, 1], \tag{15}$$

In the equation, $n$ represents the total number of engines, $\hat{y}$ represents the predicted RUL value for the $i$-th engine, and $y_i$ represents the true RUL value for the $i$-th engine.

### 4.3. Selection of Key Parameters

This paper conducted experiments using monitoring samples from four datasets, FD001 to FD004 [26,27]. Figure 6 illustrates the degradation process of the monitoring parameters for engine 32 in the FD001 dataset as the operating cycles increase, with the final lifecycle being the actual failure time of the engine. From the graph, it can be observed that some monitoring parameters, such as W31 and W32, showed a clear degradation or upward trend when increasing operating cycles, whereas others, such as T2 and P2, did not show a significant trend when increasing operating cycles.

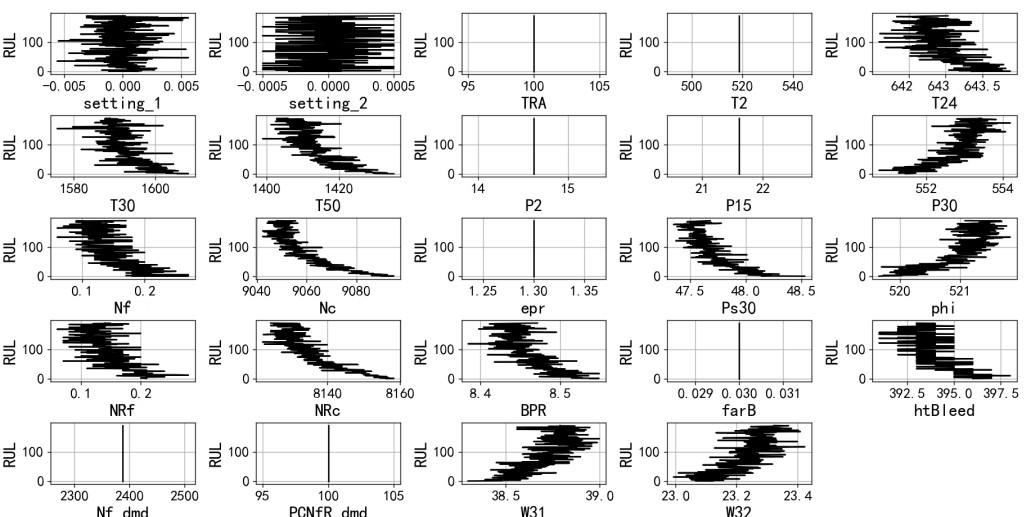

**Figure 6.** The trend of the original monitoring parameters for engine 32 in the training set FD001.

In the dataset, there are many original features, and the high correlation between many features can lead to the problem of a high dimensionality. To find the features with the highest contribution to the prediction algorithm from the massive feature data, feature selection is required. By using the RF [28] algorithm to evaluate the importance of 21 features in the original sample data, a ranking of feature variable importance was obtained. As shown in Figure 7, the importance ranking of the 21 features in the FD001 dataset is presented. These features are defined in Table 2. From Figure 7, it can be seen that the importance scores of some feature variables, such as T2, P2, Epr, FarB, Nf_dmd, PCNfR_dmd, and P15, are close to zero, indicating that these features have a weak correlation with the engine's lifecycle. Therefore, these features were removed, and the remaining 14 feature variables were considered as inputs to the prediction model.

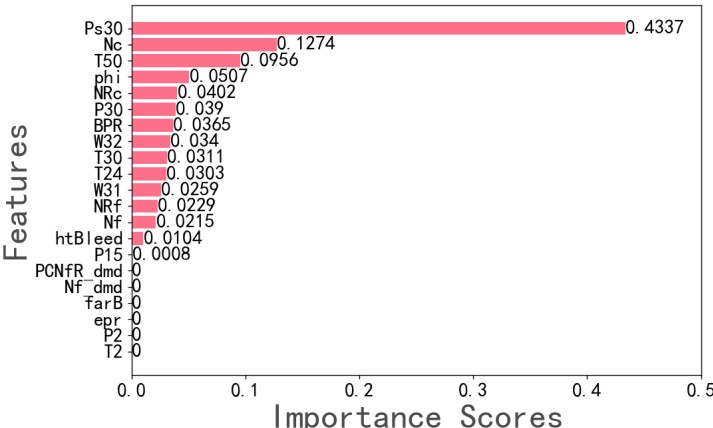

**Figure 7.** Feature variable importance ranking chart.

### 4.4. Data Processing

Due to the susceptibility of original monitoring features to noise interference, the SES algorithm was introduced to smooth [29] the monitoring information. Equation (4) was used to process the extracted features. In this paper, smoothing coefficients of 0.1 and 0.3 were selected to smooth the monitored parameters of the 14 sensors in the engine. Two randomly selected sensor parameters were plotted to show the smoothing effect before and after the smoothing process, as shown in Figure 8. The smoothing effect is the best when the coefficient is set at 0.1.

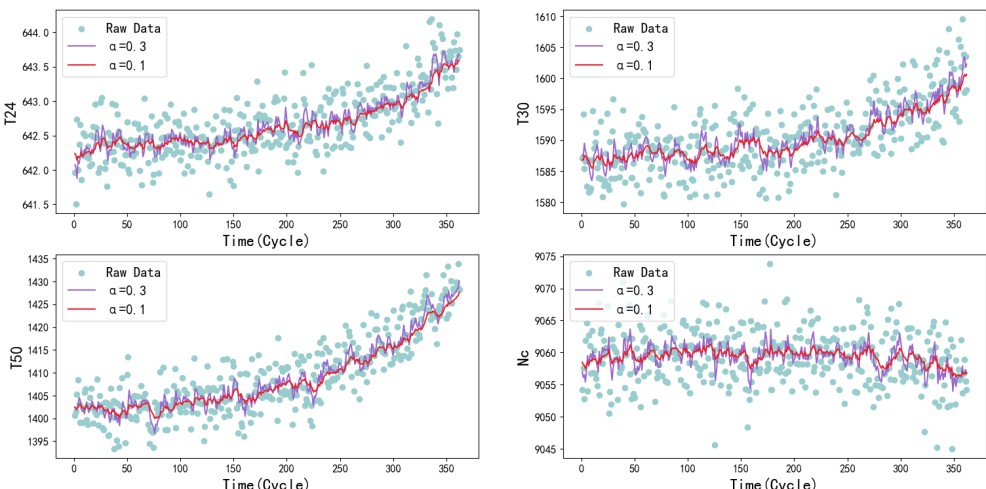

**Figure 8.** The smoothing effect of different smoothing coefficients on sensor parameters.

The numerical range of monitoring parameters in aviation engines has significant differences and the dimensions of data are not unified. In order to reduce the impact of different variable value ranges on the prediction model, the MinMax Scale [30,31] normalization method was used to map the values of the data into the [0, 1] range.

$$x_{scaled} = \frac{x - x_{min}}{x_{max} - x_{min}}, \tag{16}$$

In the formula, $x_{max}$ and $x_{min}$ represent the maximum and minimum values of each group of monitoring parameters, respectively. Standardization was performed on the 69th engine with the longest running cycle in the FD001 dataset. Before normalizing the monitoring parameters, the values of each parameter had a large range, making it difficult to see the change patterns of the monitoring parameters. The results after normalization are shown in Figure 9, where there is an obvious trend of performance degradation in the monitoring parameters.

### 4.5. Training and Optimization of RUL Prediction Model

To effectively avoid overfitting in the MLP model, L2 regularization [32] was introduced to reduce overfitting. In this study, the optimization strategies included LBFGS, Sgd, and Adam [33], which were combined with three different activation functions. The training error evaluation indicators of the model were compared using $R^2$ and the mean absolute error (MAE). Table 3 shows the comparative results of the prediction models.

Based on the results presented in Table 3, we found that the optimal choice of optimization strategy and activation function was LBFGS and ReLU, respectively, as these corresponded to coefficient of determination values closer to 1 and minimized the average absolute error. To identify the optimal structure of the MLP model, we systematically varied the number of hidden layers between 2 and 8 while setting the number of hidden nodes to 15, 25, 35, and 45, with LBFGS and ReLU [33] used as the weight optimizer and activation function, respectively. The experimental results are presented in Table 4.

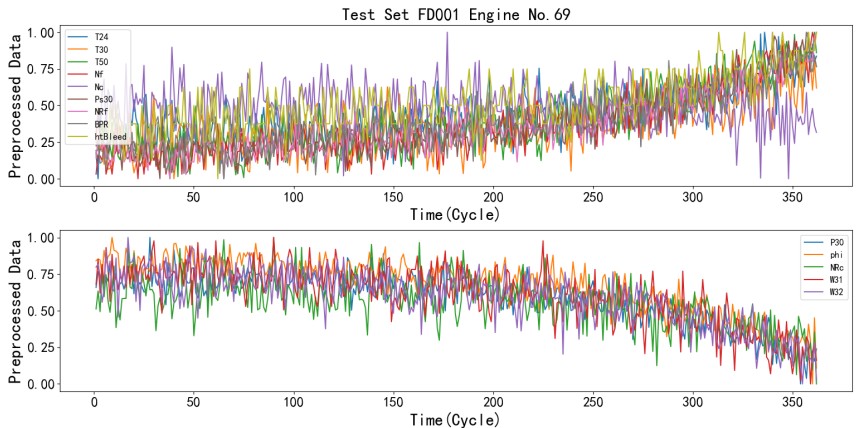

**Figure 9.** The trend of normalized monitoring parameters.

**Table 3.** Evaluation of MLP optimization strategies and activation function performance.

| | Identity | Identity | Logistic | Logistic | Tanh | Tanh | ReLU | ReLU |
|---|---|---|---|---|---|---|---|---|
| – | $R^2$ | MAE | $R^2$ | MAE | $R^2$ | MAE | $R^2$ | MAE |
| LBFGS | 0.7 | 17.93 | 0.79 | 13.94 | 0.8 | 13.62 | 0.81 | 12.98 |
| Sgd | 0.7 | 17.84 | 0.79 | 13.57 | 0.8 | 13.65 | 0.80 | 13.94 |
| Adam | 0.7 | 17.87 | 0.76 | 14.42 | 0.79 | 13.52 | 0.78 | 14.23 |

**Table 4.** Evaluation of MLP hidden layer performance.

| Performance Indicator | Number of Layers | Number of Nodes (15) | Number of Nodes (25) | Number of Nodes (35) | Number of Nodes (45) |
|---|---|---|---|---|---|
| RMSE | 2 | 17.88 | 18.99 | 19.19 | 18.39 |
| Score | 2 | 571.54 | 762.14 | 752.13 | 642.38 |
| RMSE | 4 | 19.16 | 17.70 | 17.29 | 19.04 |
| Score | 4 | 784.26 | 577 | 510.80 | 824.43 |
| RMSE | 6 | 19.19 | 17.85 | 19.44 | 18.41 |
| Score | 6 | 724.27 | 554.06 | 929.44 | 629.32 |
| RMSE | 8 | 17.92 | 18.16 | 18.61 | 19.54 |
| Score | 8 | 611.64 | 626.56 | 720.00 | 953.93 |

Figure 10 shows the validation set RMSE [34] of the MLP training model under different regularization parameters and maximum iteration numbers. From the figure, it can be observed that the RMSE is minimized when the maximum number of iterations of the MLP algorithm is set to 150 and alpha is set to 0.0001, and the model has the best training effect.

The Bayesian parameter updating algorithm was used to optimize the parameters in MLP, and the results of hyperparameter selection are shown in Table 5.

**Table 5.** Hyperparameter selection.

| Parameters | Bayesian Optimization Result |
|---|---|
| Activation | relu |
| Solver | lbfgs |
| Alpha | 0.0001 |
| Maxiter | 149 |
| Randomstate | 1 |

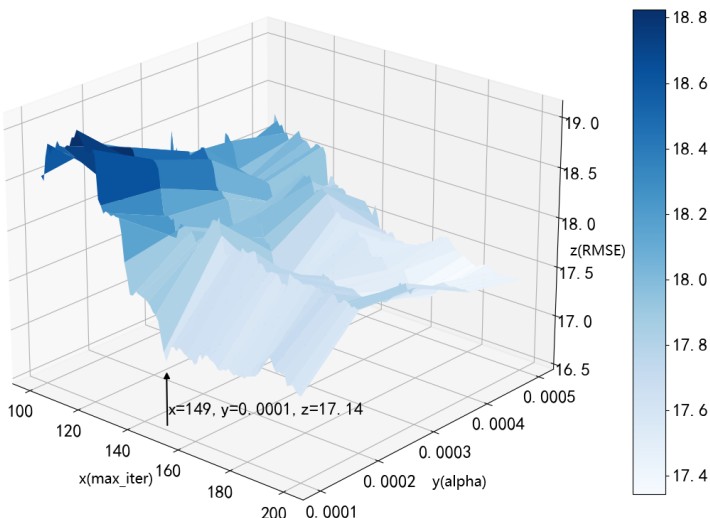

**Figure 10.** Validation set RMSE of MLP with different parameter values.

*4.6. Prediction Results*

In order to investigate the impact of the method proposed in this paper on different datasets, experiments were conducted using four datasets: FD001, FD002, FD003, and FD004. Since FD001 and FD003 only contain the life degradation information of 100 engines under a single working condition, the prediction difficulty is relatively low. In contrast, FD002 and FD004 contain historical degradation data of 248 engines under multiple working conditions, which increase the prediction difficulty of the model. Therefore, the prediction results for the FD001 and FD003 datasets are better than those for the FD002 and FD004 datasets.

*4.7. Experiment Comparison and Analysis*

There is a certain mapping relationship between the performance degradation state of each aircraft engine and its remaining life cycle. In order to explore and analyze the relationship between the engine life cycle, 81, 32, 76, and 56 randomly selected aerospace engines from the FD001 to FD004 datasets were used to predict the entire life cycle. To better evaluate the accuracy of this paper's method in predicting engine RUL, ridge regression [35], lasso regression [36], and light gradient boosting machine (LGBM) [37] methods were compared with the method proposed. As shown in Figure 11, the predicted RUL values of the aircraft engines using the method proposed are closer to the true values compared to other methods, and the predicted results are within the 95% confidence interval.

In order to further evaluate the performance of the proposed method in terms of the prediction error, four different models constructed as mentioned above were compared. The boxplots of the prediction errors are shown in Figure 12. The orange solid line in each box represents the expected value of the predicted RUL of the engine. The size of the box in the boxplot can measure the uncertainty of RUL prediction. The RUL prediction error boxplots for datasets FD002 and FD004 are relatively large compared to other test sets. This is because the data in these datasets were collected in a complex environment with multiple operating conditions, indicating that there are more uncertainty factors in the multi-operational mode. In the verification sets of FD001 and FD004 under single-operational and multi-operational conditions, engine 32 and engine 56 were randomly selected, respectively, and the probability density curves of RUL prediction values at different cycle periods are shown in Figure 13. As the cycle period increases, the RUL prediction value becomes closer to the true value.

To objectively evaluate and verify the RUL prediction results under different operating conditions, we constructed the three methods mentioned above and selected four prediction methods from the literature, including K-Neighbors Regressor (KNR) [38], to compare and analyze them with proposed method. The evaluation metrics of the RMSE and $R^2$

values of the aviation engine RUL prediction results were compared. The comparison results are shown in Figure 14. The evaluation metric RMSE for the test sets FD001 and FD003 is considerably smaller than that for FD002 and FD004 under multiple operating conditions, and the $R^2$ value is also closer to 1. Moreover, compared with the evaluation results of other methods, the RUL prediction results of our proposed method for aviation engines have mostly smaller evaluation metrics. This effectively shows that our proposed RUL prediction method has a better prediction accuracy and performance, and that the predicted RUL values of the engine are closer to the actual values.

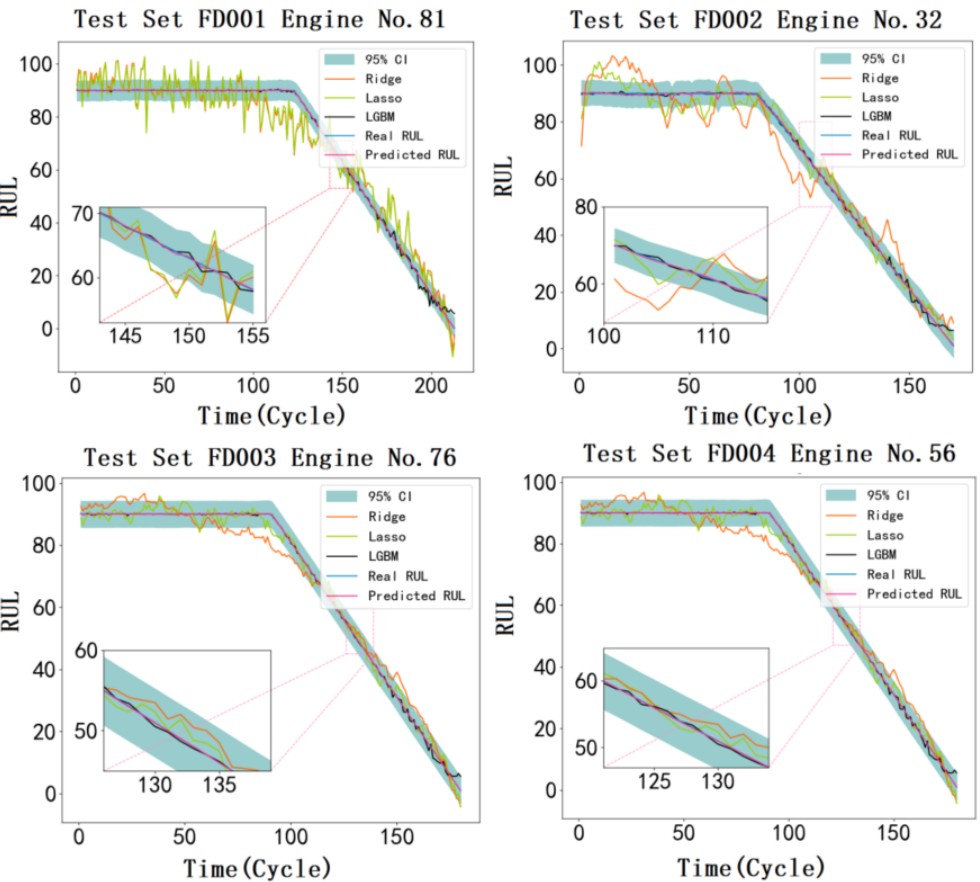

**Figure 11.** Comparing the RUL prediction results of different methods on different engines.

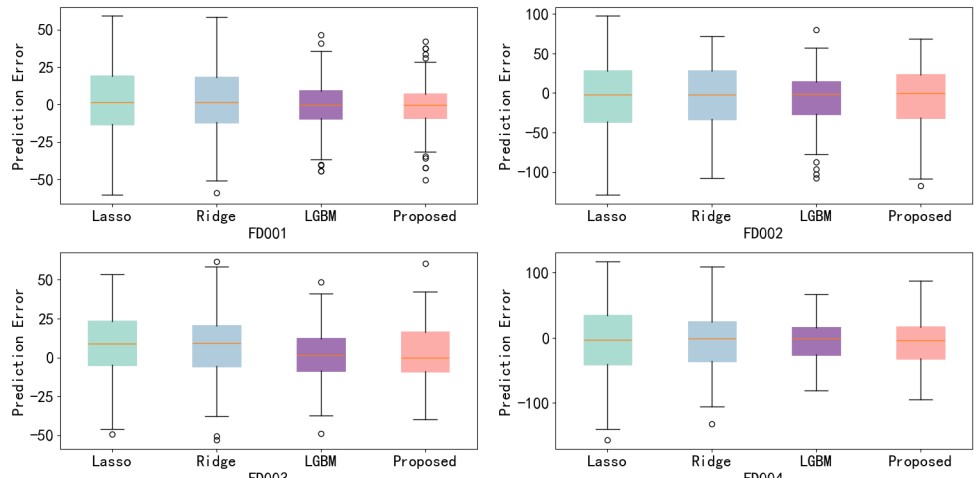

**Figure 12.** Box plots depicting prediction errors of various models on different datasets.

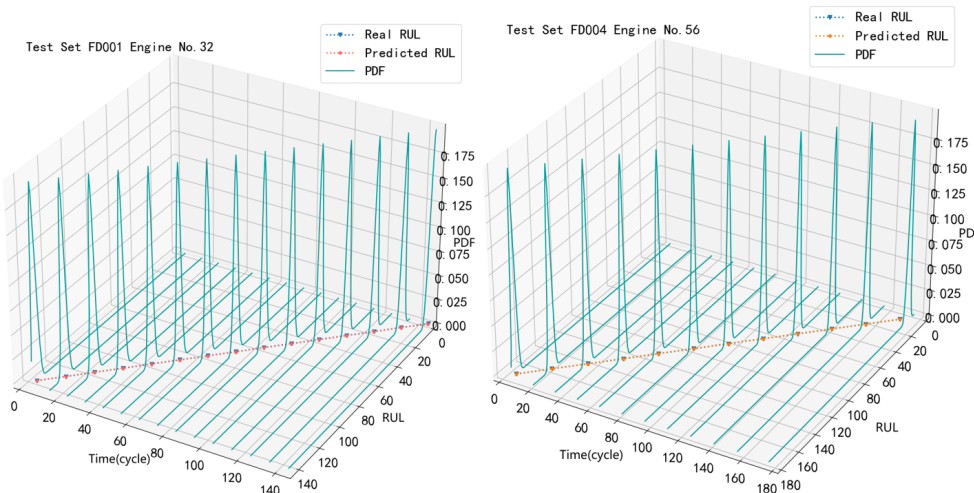

**Figure 13.** PDF and the predicted RUL values compared to the actual values.

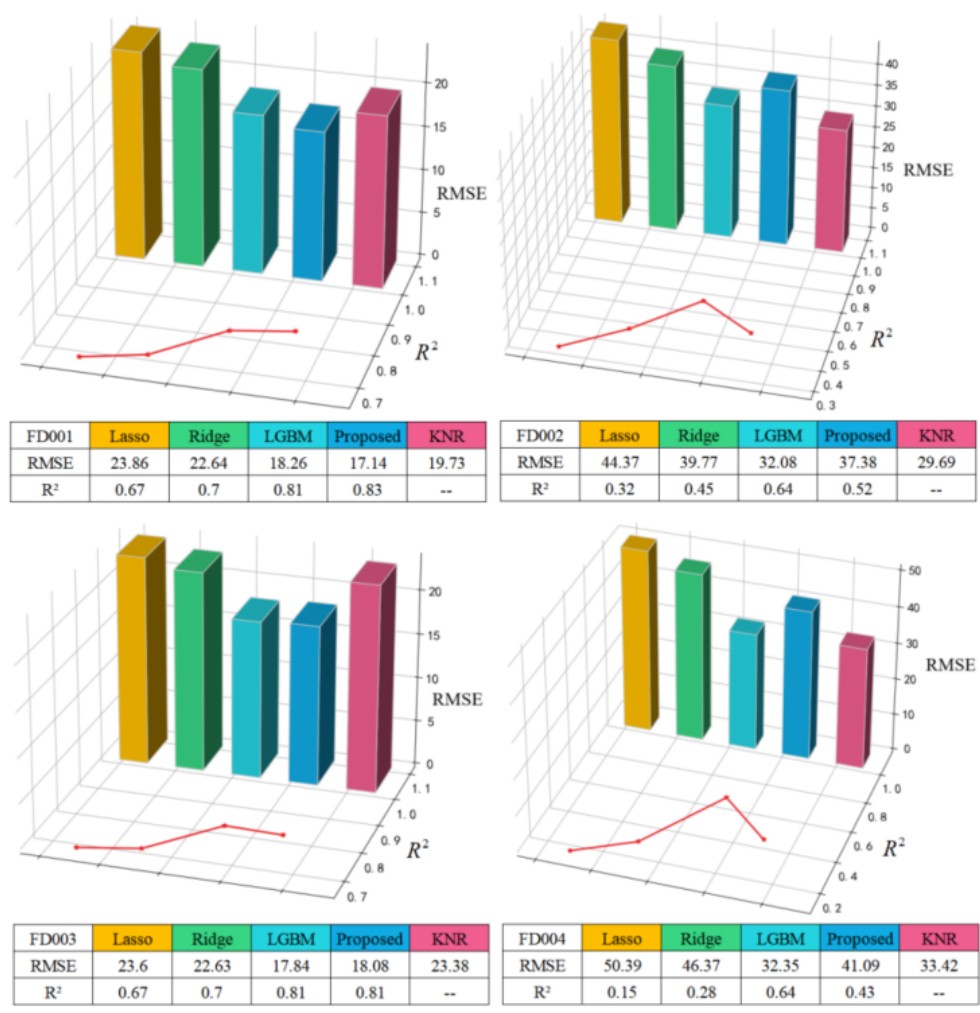

| FD001 | Lasso | Ridge | LGBM | Proposed | KNR |
|---|---|---|---|---|---|
| RMSE | 23.86 | 22.64 | 18.26 | 17.14 | 19.73 |
| R² | 0.67 | 0.7 | 0.81 | 0.83 | -- |

| FD002 | Lasso | Ridge | LGBM | Proposed | KNR |
|---|---|---|---|---|---|
| RMSE | 44.37 | 39.77 | 32.08 | 37.38 | 29.69 |
| R² | 0.32 | 0.45 | 0.64 | 0.52 | -- |

| FD003 | Lasso | Ridge | LGBM | Proposed | KNR |
|---|---|---|---|---|---|
| RMSE | 23.6 | 22.63 | 17.84 | 18.08 | 23.38 |
| R² | 0.67 | 0.7 | 0.81 | 0.81 | -- |

| FD004 | Lasso | Ridge | LGBM | Proposed | KNR |
|---|---|---|---|---|---|
| RMSE | 50.39 | 46.37 | 32.35 | 41.09 | 33.42 |
| R² | 0.15 | 0.28 | 0.64 | 0.43 | -- |

**Figure 14.** Comparison of RUL prediction results.

## 5. Conclusions

This paper investigated the problem of predicting the RUL of aircraft engines in a random degradation process under the background of complex and nonlinear dimensions of multi-source monitoring parameters. A multi-layer perceptron RUL prediction model

for aircraft engines based on the random forest and Bayesian online parameter updating was proposed. Verification and analysis were conducted using the C-MAPSS data.

In the early stage of predicting the RUL of aircraft engines, the RF algorithm was used to effectively extract features that have the greatest impact on the engine operating cycle and exhibit obvious degradation trends, thereby reducing the impact of data redundancy on the accuracy of the RUL prediction model.

In the RUL prediction stage of aircraft engines, the Bayesian parameter updating algorithm was studied to optimize the hyperparameters in the MLP. Different parameters were tested to determine their degree of influence on the RUL prediction model, obtaining the optimal parameters that minimize the evaluation index RMSE. This effectively avoids the impact of random parameter selection in the MLP prediction model on the training efficiency. After conducting experimental validation using the C-MPASS dataset, the RMSE value for the FD001 dataset decreased by 6.1% compared to other state-of-the-art methods, whereas the R-squared value increased by 2.4%.

However, there are still some shortcomings in the proposed method of this paper that are worth further improvement and refinement. Constructing a health index (HI) for the engine can reflect the health status of the engine to the greatest extent possible. Using the HI constructed as the input of the prediction model can make the RUL prediction results more accurate. Although this paper studied and modeled the aircraft engine degradation process under single and multiple working conditions, a comprehensive HI needs to be constructed based on the degradation features of the engine to characterize the change and health status of the engine during operation, and the HI needs to be further explored and analyzed in future research.

**Author Contributions:** Conceptualization, H.W. and G.Z.; methodology, D.L. (Dongwen Li); software, X.Y.; validation, D.L. (Dongwen Li), H.W., and G.Z.; formal analysis, C.L.; investigation, D.L. (Dongjun Li); resources, D.L. (Dongjun Li); data curation, C.L.; writing—original draft preparation, D.L. (Dongwen Li); writing—review and editing, H.W.; visualization, G.Z.; supervision, D.L. (Dongjun Li); project administration, H.W.; funding acquisition, H.W. and G.Z. All authors have read and agreed to the published version of the manuscript.

**Funding:** This work was supported in part by the National Natural Science Foundation of China (61863016).

**Institutional Review Board Statement:** Not applicable.

**Informed Consent Statement:** Not applicable.

**Data Availability Statement:** The datasets used in this study are openly available on NASA repository and they are called Turbofan Engine Degradation Simulation Dataset and PHM08 Challenge Dataset: (https://www.nasa.gov/intelligent-systems-division).

**Conflicts of Interest:** The authors declare no conflict of interest.

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
