# Peer review of "Remaining Useful Life Prediction of Aircraft Turbofan Engine Based on Random Forest Feature Selection and Multi-Layer Perceptron"

_applsci, doi:10.3390/app13127186_

Round 1
Reviewer 1 Report
1. What is the main question addressed by the research?
The main question of the research is Remaining Useful Life prediction stage of aircraft engienes, the Bayesian parameter updating algorithm is studied to optimize the hyperparameters in the MultyLayer Perceptron.
2. What does it add to the subject area compared with other published material?
The reaserch confirms or is similar as the another author for example authors Zhang, C., 2017 and Martha, Z., 2015.
3. Are the conclusions consistent with the evidence and arguments presented and do they address the main question posed?
The conclusion were prediction of degradation process of aircraft engine, but the resuslts investigated that comprehensive index to characterize the change and helat status of the engine during operation. Authors stated that further exploration and analysis are needed in future research. I missing discussion in the article.
4. Are the references appropriate?
The Reference are appropriate.
5. Please include any additional comments on the tables and figures.
Tables and figures are good visible.
Author Response
Dear Reviewer,
Thank you for reviewing our manuscript entitled "Remaining Useful Life Prediction of Aircraft Turbofan Engine Based on Random Forest Feature Selection and Multi-layer Perceptron". We sincerely appreciate your valuable feedback and suggestions, as they have greatly contributed to enhancing the quality and impact of our research. We have thoroughly examined each of your comments and are pleased to inform you that we have successfully addressed the required revisions. In light of your feedback, we have implemented the following revisions:
Point 1: What is the main question addressed by the research?The main question of the research is Remaining Useful Life prediction stage of aircraft engienes, the Bayesian parameter updating algorithm is studied to optimize the hyperparameters in the MultyLayer Perceptron.
Response 1: This paper mainly focuses on the problem of predicting the remaining useful life of aircraft engines. The random forest algorithm is used to extract the historical degradation features of the engine. The Bayesian parameter updating algorithm is used to update the parameters in the multilayer perceptron, and the selected features are input into the constructed model for training to obtain the remaining life of the engine.
Point 2: What does it add to the subject area compared with other published material?The reaserch confirms or is similar as the another author for example authors Zhang, C., 2017 and Martha, Z., 2015.
Response 2: Compared with other published materials, this paper fully utilizes the historical degradation data of aviation engine monitoring and extracts key degradation features that can fully characterize the engine operating cycle. Compared with reference [41] Zhang, C., 2017, the method proposed in this paper achieves lower engine RUL prediction results in FD001 and FD003 datasets, effectively improving the accuracy of prediction. This paper focuses on the nonlinear characteristics of aviation engine monitoring data, while the main core work of reference [40] Martha, Z., 2015 is to address the selection problem of linear input-output.
Point 3: Are the conclusions consistent with the evidence and arguments presented and do they address the main question posed?The conclusion were prediction of degradation process of aircraft engine, but the resuslts investigated that comprehensive index to characterize the change and helat status of the engine during operation. Authors stated that further exploration and analysis are needed in future research. I missing discussion in the article.
Response 3: According to your suggestion, we have added a discussion of related issues in the conclusion section of the article.(Line 363):
However, there are still some shortcomings in the proposed method of this paper, which are worth further improvement and refinement. Constructing a Health Index (HI) for the engine can reflect the health status of the engine to the greatest extent possible. Using the HI constructed as the input of the prediction model can make the RUL prediction results more accurate. Although this paper has studied and modeled the aircraft engine degradation process under single and multiple working conditions, a comprehensive HI needs to be constructed based on the degradation features of the engine to characterize the change and health status of the engine during operation, and the HI needs to be further explored and analyzed in future research.
Point 4: Are the references appropriate?The Reference are appropriate.
Response 4: The paper mainly cites the papers related to the prediction of residual service life of engines that are relevant to the topic of this paper. The cited references mainly include reference materials related to related research, theories, data introduction, and experiments in the field of this topic.
Point 5: Please include any additional comments on the tables and figures.Tables and figures are good visible.
Response 5: The picture content of this paper mainly includes the model structure diagram of the Random Forest algorithm and the Multilayer Perceptron algorithm, the Bayesian optimization structure diagram of MLP, the overall flow diagram of aircraft engine RUL prediction, the CMAPSS monitoring parameter description diagram, and the experimental result diagram. The content in the table describes the scale, quantity, monitoring parameters, and experimental results of the engine dataset.
We express our sincere gratitude for your positive evaluation of the manuscript's organization and publication potential, as well as your recognition of the practical significance of our research. Taking into account your valuable suggestion, we have diligently revised the manuscript to enhance its robustness and suitability for publication.
We firmly believe that these revisions have significantly elevated the overall quality and comprehensiveness of our manuscript. Once again, we extend our appreciation for your valuable feedback and the considerable time and effort you invested in reviewing our work. Your guidance has provided us with the opportunity to strengthen our research further. We are hopeful that our revised manuscript meets your expectations and eagerly await your further guidance.
Sincerely,
The Authors

Reviewer 2 Report
First of all, I would like to congratulate the authors for this interesting manuscript. I recommend accepting the manuscript considering the following minor comments.
The manuscript presents a prediction of the useful life of aircraft turbofan engines using different machine learning methods.
- - There are some undefined abbreviations such as “SEL algorithm” and “PHM method”
- - I suggest renaming the multilayer perception machine to “artificial neural networks” because it is generally recognized by this name.
- - With this high number of hidden layers and nodes, how did ensure generalization (avoid over fitting) of your model?
- - As a suggestion and in my view, figure 11 is not important and can be removed to reduce the number of figures.
Do general grammar check
Do general grammar check, specifically the definite article.
Author Response
Dear Reviewer,
Thank you for reviewing our manuscript entitled "Remaining Useful Life Prediction of Aircraft Turbofan Engine Based on Random Forest Feature Selection and Multi-layer Perceptron". We sincerely appreciate your valuable feedback and suggestions, as they have greatly contributed to enhancing the quality and impact of our research. We have thoroughly examined each of your comments and are pleased to inform you that we have successfully addressed the required revisions. In light of your feedback, we have implemented the following revisions:
Point 1: There are some undefined abbreviations such as “SEL algorithm” and “PHM method”.
Response 1: We have explained the "SEL algorithm" which was not defined in the first appearance of the abstract section in the paper (Line 8). Similarly, we defined the "PHM method" which appeared for the first time in the introduction. Based on your valuable feedback, the revised content is as follows(Line 27):
Single Exponent Smoothing(SES) algorithm is introduced for smoothing the extracted features to reduce the interference of original (Line 8)
As one of the cores of , Prognostics and Health Management (PHM) Method, aero-engine RUL prediction has been divided into (Line 27)
Point 2: I suggest renaming the multilayer perception machine to “artificial neural networks” because it is generally recognized by this name.
Response 2: We appreciate your valuable feedback. Indeed, Multilayer Perceptron is a type of artificial neural network. We have added the relationship between Multilayer Perceptron and artificial neural networks in our paper.(Line 123):
Multilayer Perceptron is a type of artificial neural networ (ANN).
Point 3: With this high number of hidden layers and nodes, how did ensure generalization (avoid over fitting) of your model?
Response 3: To tackle the overfitting issue during the model training process, we applied L2 regularization to the model to reduce overfitting and ensure the model's generalization ability.(Line 262):
To effectively avoid overfitting in the MLP model, L2 regularization was introduced to reduce overfitting.
Point 4: As a suggestion and in my view, figure 11 is not important and can be removed to reduce the number of figures.
Response 4: According to your suggestion, I have removed Figure 11 from the article and revised the relevant text.(Line 202):
To investigate the impact of the method proposed in this paper on different datasets, experiments were conducted using four datasets: FD001, FD002, FD003, and FD004. Since FD001 and FD003 contain only the life degradation information of 100 engines under a single working condition, the prediction difficulty is relatively low. In contrast, FD002 and FD004 contain historical degradation data of engines under multiple working conditions, which increases the prediction difficulty of the model. Therefore, the prediction results for FD001 and FD003 datasets are better than those for FD002 and FD004 datasets.
We express our sincere gratitude for your positive evaluation of the manuscript's organization and publication potential, as well as your recognition of the practical significance of our research. Taking into account your valuable suggestion, we have diligently revised the manuscript to enhance its robustness and suitability for publication.
We firmly believe that these revisions have significantly elevated the overall quality and comprehensiveness of our manuscript. Once again, we extend our appreciation for your valuable feedback and the considerable time and effort you invested in reviewing our work. Your guidance has provided us with the opportunity to strengthen our research further. We are hopeful that our revised manuscript meets your expectations and eagerly await your further guidance.
Sincerely,
The Authors

Reviewer 3 Report
The paper proposes a novel method to estimate RUL based on artificial intelligence, by both combining random forest and MLP. This is indeed a hot topic of research. The research behind this manuscript is sound and so is the dataset used for training data.
Have the authors tried the method in other datasets apart from C-MAPSS? I think the paper could benefit from some dataset independent tests, proving that the algorithm works for any given situation.
Finally, I have some minor concerns:
Line 6: "here. First", missing space between sentences.
Line 8: please use Single Exponent Smoothing instead of SES.
Line 26: "and safety [2].", separate citation from safety.
Line 28: "by domestic and foreign scholars", what does that mean? Wouldn't it be better to just say "by other scholars"?
Line 29: "methods [3–5].", separate citation.
Line 48: " information. In terms", missing space between sentences.
Line 54: "GoodfelleIan et al. [12]", citation 12 corresponds to Zhang et al., please correct either the text or the citation.
Line 73: "The RF algorithm", random forest? please define. Also, RF is not used on the text so consider removing it altogether.
Line 83: "using the RMSE and . " and what? missing a part of the sentence.
Line 181: "Bayes MLP", separate to be consistent with the rest of the paper.
Line 183: "Next, we", separate.
Line 201: "datasets .0is shown", fix please.
Line 211: "Performance Score Function , ", missing (Score) I think?
Line 214: "while Score penalizes" I understand Score will get defined at line 211
Line 215: Definition of "Score" should go to line 211
Line 237,238: Please be consistent with the names on table 2 for the metrics. Moreover, it would be good to remind the reader at this point that the metrics are defined on table 2.
Line 243: "Formula (4) is", please use equation instead of formula and use a proper reference for equation 4.
Line 263: " LBFGS, Sgd, and Adam, ", should need a citation.
Line 265: "R-squared", is this the same metric than R^2 defined at equation 15? If so then use the same symbol. If not please define it.
Line 268: " LBFGS and ReLU,", should need a citation.
Line 276: "root mean square error", use RMSE.
Line 276: What is Maxiter? Maybe something like "the maximum number of iterations of the .... algorithm is".
Lines 295,296: "Ridge 295 regression, Lasso regression, Light Gradient Boosting Machine (LGBM) methods", should require a citation of these methods.
Line 315: "KNR [41]", separate citation from text. Also what is KNR, please define.
Line 320: "to 1. Moreover,", please separate sentences.
Author Response
Dear Reviewer,
Thank you for reviewing our manuscript entitled "Remaining Useful Life Prediction of Aircraft Turbofan Engine Based on Random Forest Feature Selection and Multi-layer Perceptron". We sincerely appreciate your valuable feedback and suggestions, as they have greatly contributed to enhancing the quality and impact of our research. We have thoroughly examined each of your comments and are pleased to inform you that we have successfully addressed the required revisions. In light of your feedback, we have implemented the following revisions:
Point 1: Have the authors tried the method in other datasets apart from C-MAPSS? I think the paper could benefit from some dataset independent tests, proving that the algorithm works for any given situation.
Response 1: We appreciate your valuable feedback. In this paper, we only conducted experiments on the C-MAPSS dataset and did not use other datasets to test our method. In future research, we will continue to try other datasets and compare them with the C-MAPSS dataset to demonstrate the applicability of this algorithm.
Point 2: Line 6: "here. First", missing space between sentences.
Response 2: We appreciate your correction to the paper, and we have made the revision to add a space between "here." and "First" in the sentence (Line 5):
and Bayes-optimized multilayer perceptron (MLP) is proposed here. First
Point 3: Line 8: please use Single Exponent Smoothing instead of SES.
Response 3: We appreciate your correction to the paper, and we have revised "SES" in the paper to "Single Exponential Smoothing (SES)". (Line 8):
Then, the Single Exponent Smoothing (SES) algorithm is introduced for smoothing the
Point 4: Line 26: "and safety [2].", separate citation from safety.
Response 4: In line 26, we have separated the citation from the sentence and changed "and safety[2]." to "and safety [2]."(Line 26):
equipment reliability and safety [2].
Point 5: Line 28: "by domestic and foreign scholars", what does that mean? Wouldn't it be better to just say "by other scholars"?
Response 5:The phrase "by domestic and foreign scholars" means scholars from both home and abroad. Based on your feedback, we have revised line 28 of the paper to "by other scholars".(Line 28):
aero-engine RUL prediction has been divided into three main approaches by other scholars
Point 6: Line 29: "methods [3–5].", separate citation.
Response 6: Based on your feedback, we have revised line 29 by adding separate citations for "methods [3-5]." The updated version is as follows.(Line 29):
physical model-based methods[3], data-driven methods[4], and hybrid methods[5].
Point 7: Line 48: " information. In terms", missing space between sentences.
Response 7: Based on your feedback, we have added a space between "information" and "In terms" in the sentence.(Line 49):
on identifying the most valuable information. In terms information.In terms of feature
Point 8: Line 54: "GoodfelleIan et al. [12]", citation 12 corresponds to Zhang et al., please correct either the text or the citation.
Response 8: We appreciate your correction on the article. As there was a mistake in the citation of [12], we have now corrected it as follows: (Line 54):
Zhang et al. [12] proposed a method for predicting remaining useful life of engines using a combination of convolutional and recurrent neural networks (CNN-RNN), and the prediction accuracy is further improved by processing and clustering the data.
Point 9: Line 73: "The RF algorithm", random forest? please define. Also, RF is not used on the text so consider removing it altogether.
Response 9: "The RF algorithm" refers to the Random Forest algorithm used for feature extraction in this article. Based on your suggestion, I have changed "The RF algorithm" to "The Random Forest algorithm". (Line 77):
Point 10: Line 83: "using the RMSE and . " and what? missing a part of the sentence.
Response 10: Thank you for pointing out the mistake in my paper. Due to my negligence, R2 was omitted and it should be corrected as "using the RMSE and R2."(Line 87)
and the RUL prediction results are evaluated using the RMSE and R2.
Point 11: Line 181: "Bayes MLP", separate to be consistent with the rest of the paper.
Response 11: Based on your feedback, we have changed "BayesMLP" to "Bayes MLP" in the paper. (Line 186):
We begin by pre-processing the test set, which is then input into the Bayes MLP
Point 12: Line 183: "Next, we", separate.
Response12 : Based on your feedback, we have changed "Next,we" to "Next,we" in the paper. (Line 189):
Next, we calculate the expectation and standard deviation of the RUL prediction
Point 13: Line 201: "datasets .0is shown", fix please.
Response 13: Based on your suggestion, we have changed "datasets .0is shown" to "datasets FD001-FD004 is shown" in the paper.(Line 206):
The maximum cycle sample length distribution of all engines in datasets FD001-FD004 is shown in Figure 5.
Point 14: Line 211: "Performance Score Function ", missing (Score) I think?
Response 14: Based on your suggestion, we have changed "Performance Score Function" to "Performance Score Function (Score)" in the paper.(Line 217):
and Performance Score Function (Score) are introduced to assess the model’s performance
Point 15: Line 214: "while Score penalizes" I understand Score will get defined at line 211
Response 15: Indeed, we have added back "(Score)" in line 211.(Line 217)
and Performance Score Function (Score) are introduced to assess the model’s performance
Point 16: Line 215: Definition of "Score" should go to line 211
Response 16:Based on your feedback, we have added the definition of "Score" to line 211 and deleted line 215.(Line 217 and 221)
Point 17: Line 237,238: Please be consistent with the names on table 2 for the metrics. Moreover, it would be good to remind the reader at this point that the metrics are defined on table 2.
Response 17: We have modified the descriptions of the defined metrics to be consistent with those in Table 2. In addition, we have included a reminder in the paper that the definitions of these metrics can be found in Table 2.(Line 243)
These features are defined in Table 2 . From the figure 7, it can be seen that the importance scores of some feature variables, such as T2, P2, Epr, FarB, Nf_dmd, PCNfR_dmd, and P15 are close to zero,
Point 18: Line 243: "Formula (4) is", please use equation instead of formula and use a proper reference for equation 4.
Response 18: Based on your suggestion, we have changed "Formula (4) is" to "Equation (4) is" and included a reference [16] in the text for the fourth equation.(Line 252):
Equation (4) [16] is used to process the extracted features.
Point 19: Line 263: " LBFGS, Sgd, and Adam, ", should need a citation.
Response 19: According to your suggestion, we have cited "LBFGS, Sgd, and Adam," and added a reference [42] to it.(Line 273):
In this study, the optimization strategies included LBFGS, Sgd, and Adam [42],
Point 20: Line 265: "R-sï¼›quared", is this the same metric than R^2 defined at equation 15? If so then use the same symbol. If not please define it.
Response 20: The "R-squared" metric in the text refers to the same metric as "R2". Based on your suggestion, we have changed "R-squared" to "R2" in the text.(Line 275):
the model were compared using R2 and mean absolute error (MAE).
Point 21: Line 268: " LBFGS and ReLU,", should need a citation.:
Response 21: According to your suggestion, we have cited "LBFGS and ReLU," and added a reference [42] to it.(Line 282):
nodes to 15, 25, 35, and 45, with LBFGS and ReLU [42],
Point 22: Line 276: "root mean square error", use RMSE.
Response 22: Based on your suggestion, we have changed "root mean square error" to "RMSE" in line 276.(Line 286):
observed that the root RMSE is minimized when Maxiter is set to 150 and alpha
Point 23: Line 276: What is Maxiter? Maybe something like "the maximum number of iterations of the .... algorithm is".
Response 23: Based on your suggestion, we have changed "Maxiter" to "the maximum number of iterations of the MLP algorithm is" in the text.(Line 286):
the RMSE is minimized when the maximum number of iterations of the MLP algorithm
Point 24: Lines 295,296: "Ridge 295 regression, Lasso regression, Light Gradient Boosting Machine (LGBM) methods", should require a citation of these methods.
Response 24: Based on your feedback, we have added references to these methods.(Line 313):
the Ridge regression [43], Lasso regression [44], Light Gradient Boosting Machine (LGBM) [45]
Point 25: Line 315: "KNR [41]", separate citation from text. Also what is KNR, please define.
Response 25: Based on your feedback, we have separated the references from the text, and KNR refers to the K-Nearest Neighbor algorithm. We have added an explanation of KNR in the text (Line 328).
methods from literature, including KNeighbors Regressor (KNR) [41], to compare
Point 26: Line 320: "to 1. Moreover,", please separate sentences.
Response 26: Based on your suggestion, we have changed "to 1. Moreover," to "to 1. Moreover," in the text. (Line 340).
and the R2 value is also closer to 1. Moreover, to 1.Moreover, compared with the evaluation
We express our sincere gratitude for your positive evaluation of the manuscript's organization and publication potential, as well as your recognition of the practical significance of our research. Taking into account your valuable suggestion, we have diligently revised the manuscript to enhance its robustness and suitability for publication.
We firmly believe that these revisions have significantly elevated the overall quality and comprehensiveness of our manuscript. Once again, we extend our appreciation for your valuable feedback and the considerable time and effort you invested in reviewing our work. Your guidance has provided us with the opportunity to strengthen our research further. We are hopeful that our revised manuscript meets your expectations and eagerly await your further guidance.
Sincerely,
The Authors
